# Differentially Private SGDA for Minimax Problems

**Zhenhuan Yang**[1]     **Shu Hu**[2]     **Yunwen Lei**[3]     **Kush R Varshney**[4]     **Siwei Lyu**[2]     **Yiming Ying**[1]

[1]University at Albany, Albany, New York, USA
[2]University at Buffalo, Buffalo, New York, USA
[3]University of Birmingham, Birmingham, UK
[4]IBM Research, Yorktown Heights, New York, USA

## Abstract

Stochastic gradient descent ascent (SGDA) and its variants have been the workhorse for solving minimax problems. However, in contrast to the well-studied stochastic gradient descent (SGD) with differential privacy (DP) constraints, there is little work on understanding the generalization (utility) of SGDA with DP constraints. In this paper, we use the algorithmic stability approach to establish the generalization (utility) of DP-SGDA in different settings. In particular, for the convex-concave setting, we prove that the DP-SGDA can achieve an optimal utility rate in terms of the weak primal-dual population risk in both smooth and non-smooth cases. To our best knowledge, this is the first-ever-known result for DP-SGDA in the non-smooth case. We further provide its utility analysis in the nonconvex-strongly-concave setting which is the first-ever-known result in terms of the primal population risk. The convergence and generalization results for this nonconvex setting are new even in the non-private setting. Finally, numerical experiments are conducted to demonstrate the effectiveness of DP-SGDA for both convex and nonconvex cases.

## 1 INTRODUCTION

In recent years, there is a growing interest on studying the minimax problems which involve both minimization over the primal variable $\mathbf{w}$ and maximization over the dual variable $\mathbf{v}$. Notable examples include generative adversarial networks (GANs) [Goodfellow et al., 2014, Arjovsky et al., 2017], AUC maximization [Gao et al., 2013, Ying et al., 2016, Natole et al., 2018, Liu et al., 2020, Zhao et al., 2011], robust learning [Audibert and Catoni, 2011, Xu et al., 2009], adversarial training [Sinha et al., 2017], algorithmic fairness

[Mohri et al., 2019, Li et al., 2019, Wang et al., 2020b, Martinez et al., 2020, Diana et al., 2021], and Markov Decision Process (MDP) [Puterman, 2014, Wang, 2017]. Details of these motivating examples are given in Appendix A.

The minimax problem can be formulated as

$$\min_{\mathbf{w} \in \mathcal{W}} \max_{\mathbf{v} \in \mathcal{V}} \left\{ F(\mathbf{w}, \mathbf{v}) := \mathbb{E}_{\mathbf{z} \sim \mathcal{D}}[f(\mathbf{w}, \mathbf{v}; \mathbf{z})] \right\}, \quad (1)$$

where $\mathcal{W} \subseteq \mathbb{R}^{d_1}$ and $\mathcal{V} \subseteq \mathbb{R}^{d_2}$ are two nonempty closed and convex domains and $\mathbf{z}$ is a random variable from some distribution $\mathcal{D}$ taking values in $\mathcal{Z}$. Since the distribution $\mathcal{D}$ is usually unknown and one has access only to an i.i.d. training dataset $S = \{\mathbf{z}_1, \cdots, \mathbf{z}_n\}$, one resorts to solving its empirical minimax problem

$$\min_{\mathbf{w} \in \mathcal{W}} \max_{\mathbf{v} \in \mathcal{V}} \left\{ F_S(\mathbf{w}, \mathbf{v}) := \frac{1}{n} \sum_{i=1}^{n} f(\mathbf{w}, \mathbf{v}; \mathbf{z}_i) \right\}.$$

One popular optimization algorithm for solving this problem is SGDA. Specifically, at iteration $t$, upon receiving a random data point or mini-batch from $S$, it performs gradient descent over $\mathbf{w}$ with the stepsize $\eta_{\mathbf{w},t}$ and gradient ascent over $\mathbf{v}$ with the stepsize $\eta_{\mathbf{v},t}$.

As SGDA is conceptually simple and easy to implement, it is widely deployed in solving minimax problems, e.g., GANs [Goodfellow et al., 2014], adversarial learning [Sinha et al., 2017], and AUC maximization [Ying et al., 2016]. Its local convergence analysis for nonconvex-(strongly-)concave problems was established in Lin et al. [2020]. Other variants of SGDA were proposed and studied in Luo et al. [2020], Nouiehed et al. [2019], Rafique et al. [2021], Yan et al. [2020].

On another front, collected data often contain sensitive information such as individual records from hospitals, online behavior from social media, and genomic data from cancer diagnosis. Differential privacy [Dwork et al., 2014] has emerged as a well-accepted mathematical definition of privacy which ensures that an attacker gets roughly the same

*Accepted for the 38th Conference on Uncertainty in Artificial Intelligence* (UAI 2022).

information from the dataset regardless of whether an individual is present or not. Its related technologies have been adopted by Google [Erlingsson et al., 2014], Apple [Ding et al., 2017], and the US Census Bureau [Abowd, 2016]. While SGD and SGDA have become the workhorse behind the remarkable progress of machine learning and AI, it is of pivotal importance for developing their counterparts with DP constraints.

Many studies analyze the privacy and utility of DP-SGD for the ERM problem that only involves the minimization over $\mathbf{w}$ [Bassily et al., 2019, 2020, Feldman et al., 2020, Song et al., 2013, Wang et al., 2021a, 2020a, 2019b, Wu et al., 2017, Zhou et al., 2020]. In contrast, there is little work on analysing the utility of minimax optimization algorithms with DP constraints except the recent work of Boob and Guzmán [2021]. However, Boob and Guzmán [2021] focus on the noisy stochastic extragradient method on convex-concave and smooth settings.

Studying the computational and statistical behavior of DP-SGDA is fundamental towards the understanding of stochastic optimization algorithm for minimax problem under the differential privacy constraint. In this paper, we propose novel convergence and stability analysis to establish the utility of DP-SGDA in empirical saddle point and population forms such as the weak primal-dual population risk and the primal population risk. We collect in Table 1 the notations and results of performance measures in this paper. In particular, our contributions can be summarized as follows.

- We analyze the privacy and utility of DP-SGDA under the convex-concave setting in terms of the weak primal-dual population risk, i.e., $\max_{\mathbf{v} \in \mathcal{V}} \mathbb{E}\big[F(A_{\mathbf{w}}(S), \mathbf{v})\big] - \min_{\mathbf{w} \in \mathcal{W}} \mathbb{E}\big[F(\mathbf{w}, A_{\mathbf{v}}(S))\big]$, where $(A_{\mathbf{w}}(S), A_{\mathbf{v}}(S))$ is the output of DP-SGDA. Specifically, we show that it can guarantee $(\epsilon, \delta)$-DP and achieve the optimal rate $\mathcal{O}\big(\frac{1}{\sqrt{n}} + \frac{\sqrt{d \log(1/\delta)}}{n\epsilon}\big)$ for smooth and nonsmooth cases where $d = \max\{d_1, d_2\}$. To our best knowledge, this is the first-ever known result for DP-SGDA in the nonsmooth case.

- We further study the utility of DP-SGDA in the nonconvex-strongly-concave case in terms of the primal population risk, i.e., $R(A_{\mathbf{w}}(S)) = \max_{\mathbf{v} \in \mathcal{V}} \mathbb{E}\big[F(A_{\mathbf{w}}(S), \mathbf{v})\big]$. In particular, under the Polyak-Łojasiewicz (PL) condition of $F_S$, we prove that the excess primal population risk, i.e., $R(A_{\mathbf{w}}(S)) - \min_{\mathbf{w} \in \mathcal{W}} R(\mathbf{w})$, enjoys the rate $\mathcal{O}\big(\frac{1}{n^{1/3}} + \frac{\sqrt{d \log(1/\delta)}}{n^{5/6}\epsilon}\big)$ while guaranteeing $(\epsilon, \delta)$-DP. The key techniques involve the convergence analysis of $R_S(A_{\mathbf{w}}(S)) - \min_{\mathbf{w}} R_S(\mathbf{w})$ and the stability analysis for $A_{\mathbf{w}}(S)$ which are of interest in their own rights. As far as we are aware, these results are the first ones known for DP-SGDA in the nonconvex setting.

- We perform numerical experiments on three benchmark

datasets which validate the effectiveness of DP-SGDA for both convex and non-convex cases.

## 1.1 MOTIVATING EXAMPLES

We give two examples of minimax problems under the DP constraint. See Appendix A for more examples and details.

**AUC Maximization.** Area Under the ROC Curve (AUC) is a widely used measure for binary classification. It has been shown optimizing AUC is equivalent to a minimax problem once auxiliary variables $a, b, v \in \mathbb{R}$ are introduced [Ying et al., 2016].

$$\min_{\theta,a,b} \max_{v} \Big\{ F(\theta, a, b, v) = \mathbb{E}_{\mathbf{z}}[f(\theta, a, b, v; \mathbf{z})] \Big\}.$$

Differential privacy has been applied to learn private classifier by optimizing AUC [Wang et al., 2021b].

**Generative Adversarial Networks.** Originally proposed in Goodfellow et al. [2014], GAN in general can be written as a minimax problem between a generator network $G_{\mathbf{v}}$ and a discriminator network $D_{\mathbf{w}}$

$$\min_{\mathbf{w}} \max_{\mathbf{v}} \mathbb{E}[f(\mathbf{w}, \mathbf{v}; \mathbf{z}, \xi)] = \mathbb{E}_{\mathbf{z}}[D_{\mathbf{w}}(\mathbf{z})] - \mathbb{E}_{\xi}[D_{\mathbf{w}}(G_{\mathbf{v}}(\xi))].$$

DP-SGDA and its variants were employed to train differential private GANs by Xie et al. [2018]. Recently differential privacy has successfully applied to private data generation by GAN framework [Jordon et al., 2018, Beaulieu-Jones et al., 2019].

## 1.2 RELATED WORK

Below we briefly discuss some related work.

**Convergence analysis for SGDA.** It is a classical result that SGDA can achieve a convergence rate $\mathcal{O}(1/\sqrt{T})$ in the convex and concave case [Nedić and Ozdaglar, 2009, Nemirovski et al., 2009] where $T$ is the number of iterations. For the nonconvex-(strongly)-concave case, the work of Lin et al. [2020] shows the local convergence of SGDA if the stepsizes $\eta_{\mathbf{w},t}$ and $\eta_{\mathbf{v},t}$ are chosen to be appropriately different. Other important studies consider variants of SGDA and prove their local convergence for the nonconvex case. Such algorithms include nested algorithms [Rafique et al., 2021] for weakly-convex-weakly-concave problems, multi-step GDA [Nouiehed et al., 2019] under the one-sided PL condition, epoch-wise SGDA [Yan et al., 2020], and stochastic recursive SGDA [Luo et al., 2020] for nonconvex-strongly-concave problems, to mention but a few.

**Stability and generalization of non-private SGD and SGDA.** The studies of [Hardt et al., 2016, Charles and Papailiopoulos, 2018, Kuzborskij and Lampert, 2018] use uniform stability Bousquet and Elisseeff [2002] to derive the generalization of non-private SGD for the convex and

| Algorithm | Assumption | Measure | Rate | Complexity | Simplicity |
|---|---|---|---|---|---|
| NSEG | C-C, Lip, S | $\triangle^w(A_{\mathbf{w}}(S), A_{\mathbf{v}}(S))$ | $\mathcal{O}\left(\frac{1}{\sqrt{n}} + \frac{\sqrt{d\log(1/\delta)}}{n\epsilon}\right)$ | $\mathcal{O}(n^2)$ | Single-loop |
| NISPP | C-C, Lip, S | $\triangle^w(A_{\mathbf{w}}(S), A_{\mathbf{v}}(S))$ | $\mathcal{O}\left(\frac{1}{\sqrt{n}} + \frac{\sqrt{d\log(1/\delta)}}{n\epsilon}\right)$ | $\mathcal{O}(n^{3/2}\log(n))$ | Double-loop |
| DP-SGDA (Ours) | C-C, Lip, S | $\triangle^w(A_{\mathbf{w}}(S), A_{\mathbf{v}}(S))$ | $\mathcal{O}\left(\frac{1}{\sqrt{n}} + \frac{\sqrt{d\log(1/\delta)}}{n\epsilon}\right)$ | $\mathcal{O}(n^{3/2})$ | Single-loop |
|  | C-C, Lip | $\triangle^w(A_{\mathbf{w}}(S), A_{\mathbf{v}}(S))$ | $\mathcal{O}\left(\frac{1}{\sqrt{n}} + \frac{\sqrt{d\log(1/\delta)}}{n\epsilon}\right)$ | $\mathcal{O}(n^{5/2})$ |  |
|  | PL-SC, Lip, S | $R(A_{\mathbf{w}}(S)) - \min_{\mathbf{w}} R(\mathbf{w})$ | $\mathcal{O}\left(\frac{1}{n^{1/3}} + \frac{\sqrt{d\log(1/\delta)}}{n^{5/6}\epsilon}\right)$ | $\mathcal{O}(n^{3/2})$ |  |

Table 1: *Summary of Results. DP-SGDA is Algorithm 1 in this paper. NSEG and NISPP are Algorithm 1 and 2 in Boob and Guzmán [2021], respectively. Here C-C means convexity and concavity, PL-SC means PL condition and strong concavity, Lip means Lipschitz continuity, S means the smoothness. $\triangle^w(A_{\mathbf{w}}(S), A_{\mathbf{v}}(S))$ is the weak PD population risk and $R(A_{\mathbf{w}}(S)) - \min_{\mathbf{w}} R(\mathbf{w})$ is the excess primal population risk.*

smooth case while the convex and nonsmooth case was established by Bassily et al. [2020], Lei and Ying [2020]. The nonconvex case under the PL-condition was considered by Charles and Papailiopoulos [2018], Lei and Ying [2021]. The stability and generalization of SGDA for minimax problems were studied by Lei et al. [2021] in different forms for convex and nonconvex, smooth, and nonsmooth cases, and by Farnia and Ozdaglar [2021] with focus on the smooth cases.

**DP-SGD and DP-SGDA.** DP-SGD was shown to attain the optimal excess population risk $\mathcal{O}(1/\sqrt{n} + \sqrt{d\log(1/\delta)}/n\epsilon)$ in Bassily et al. [2019, 2020], Wang et al. [2021a, 2020a] for the convex case. For nonconvex objectives, Wang et al. [2019a] studied the DP Gradient Langevin Dynamics, and Zhang et al. [2021b] studied a multi-stage type of DP-SGD assuming the weakly-quasi-convexity and PL condition. In Xie et al. [2018], Zhang et al. [2018], DP-SGDA and its variants together with clipping techniques were employed to train differentially private GANs which showed promising results in applications. However, no utility analysis was given there. Boob and Guzmán [2021] focused on the noisy stochastic extragradient method with DP constraints for minimax problems in the convex-concave and smooth settings and provided its utility analysis using variational inequality (VI) and stability approaches.

## 2 PROBLEM FORMULATION

In this section, we introduce necessary assumptions, notations and the DP-SGDA algorithm.

### 2.1 ASSUMPTIONS AND NOTATIONS

Firstly, we introduce necessary assumptions and notations. A function $h : \mathcal{W} \to \mathbb{R}$ is said to be convex if, for all $\mathbf{w}, \mathbf{w}' \in \mathcal{W}$, there holds $h(\mathbf{w}) \geq h(\mathbf{w}') + \langle \nabla h(\mathbf{w}'), \mathbf{w} - \mathbf{w}' \rangle$ where $\nabla$ is the gradient operator and $\langle \cdot, \cdot \rangle$ is the inner product. Let $\| \cdot \|_2$ denote the Euclidean norm. We say $h$ is

$\rho$-strongly-convex if $h - \frac{\rho}{2}\|\mathbf{w}\|_2^2$ is convex, $h$ is concave if $-h$ is convex, and $\rho$-strongly-concave if $-h - \frac{\rho}{2}\|\mathbf{w}\|_2^2$ is convex. Let $[n] := \{1, 2, \ldots, n\}$.

**Definition 1.** *Given a function $h : \mathcal{W} \times \mathcal{V} \to \mathbb{R}$. We say $h$ is convex-concave if for any $\mathbf{v} \in \mathcal{V}$, the function $\mathbf{w} \mapsto h(\mathbf{w}, \mathbf{v})$ is convex and for any $\mathbf{w} \in \mathcal{W}$, the function $\mathbf{v} \mapsto h(\mathbf{w}, \mathbf{v})$ is concave.*

**Assumption 1 (A1).** *The function $f$ is said to be Lipschitz continuous if there exist $G_{\mathbf{w}}, G_{\mathbf{v}} > 0$ such that, for any $\mathbf{w}, \mathbf{w}' \in \mathcal{W}, \mathbf{v}, \mathbf{v}' \in \mathcal{V}$ and $\mathbf{z} \in \mathcal{Z}$, $\|f(\mathbf{w}, \mathbf{v}; \mathbf{z}) - f(\mathbf{w}', \mathbf{v}; \mathbf{z})\|_2 \leq G_{\mathbf{w}}\|\mathbf{w} - \mathbf{w}'\|_2$, and $\|f(\mathbf{w}, \mathbf{v}; \mathbf{z}) - f(\mathbf{w}, \mathbf{v}'; \mathbf{z})\|_2 \leq G_{\mathbf{v}}\|\mathbf{v} - \mathbf{v}'\|_2$. And denote $G = \max\{G_{\mathbf{w}}, G_{\mathbf{v}}\}$.*

**Assumption 2 (A2).** *For randomly drawn $j \in [n]$, the gradients $\nabla_{\mathbf{w}} f(\mathbf{w}, \mathbf{v}; \mathbf{z}_j)$ and $\nabla_{\mathbf{v}} f(\mathbf{w}, \mathbf{v}; \mathbf{z}_j)$ have bounded variances $B_{\mathbf{w}}$ and $B_{\mathbf{v}}$ respectively. And let $B = \max\{B_{\mathbf{w}}, B_{\mathbf{v}}\}$.*

**Assumption 3 (A3).** *The function $f$ is said to be smooth if it is continuously differentiable and there exists a constant $L > 0$ such that for any $\mathbf{w}, \mathbf{w}' \in \mathcal{W}, \mathbf{v}, \mathbf{v}' \in \mathcal{V}$ and $\mathbf{z} \in \mathcal{Z}$,*

$$\left\| \begin{pmatrix} \nabla_{\mathbf{w}} f(\mathbf{w}, \mathbf{v}; \mathbf{z}) - \nabla_{\mathbf{w}} f(\mathbf{w}', \mathbf{v}'; \mathbf{z}) \\ \nabla_{\mathbf{v}} f(\mathbf{w}, \mathbf{v}; \mathbf{z}) - \nabla_{\mathbf{v}} f(\mathbf{w}', \mathbf{v}'; \mathbf{z}) \end{pmatrix} \right\|_2 \leq L \left\| \begin{pmatrix} \mathbf{w} - \mathbf{w}' \\ \mathbf{v} - \mathbf{v}' \end{pmatrix} \right\|_2$$

We also require the Polyak-Łojasiewicz (PL) condition.

**Definition 2** ([Polyak, 1964]). *A function $h : \mathcal{W} \to \mathbb{R}$ satisfies the PL condition if there exist a constant $\mu > 0$ such that, for any $\mathbf{w} \in \mathcal{W}$, $\frac{1}{2}\|\nabla h(\mathbf{w})\|_2^2 \geq \mu(h(\mathbf{w}) - \min_{\mathbf{w}' \in \mathcal{W}} h(\mathbf{w}'))$.*

We refer to Karimi et al. [2016] for a nice discussion of this condition and other general conditions that allow the global convergence of gradient descent.

### 2.2 DP-SGDA ALGORITHM

We now move on to the definition of differential privacy and the description of DP-SGDA. Differential privacy was

introduced by Dwork et al. [2006, 2014]. We say that two datasets $S, S'$ are neighboring datasets if they differ by at most one example.

---

**Algorithm 1** Differentially Private Stochastic Gradient Descent Ascent (DP-SGDA) Method

---

1: **Inputs:** data $S = \{\mathbf{z}_i : i \in [n]\}$, privacy budget $\epsilon, \delta$, number of iterations $T$, learning rates $\{\eta_{\mathbf{w},t}, \eta_{\mathbf{v},t}\}_{t=1}^{T}$, and initialize $(\mathbf{w}_0, \mathbf{v}_0)$
2: Compute noise parameters $\sigma_{\mathbf{w}}$ and $\sigma_{\mathbf{v}}$ based on Eq. (3)
3: **for** $t = 1$ to $T$ **do**
4:     Sample a mini-batch $I_t = \{i_t^1, \cdots, i_t^m \in [n]\}$ uniformly with replacement
5:     Sample independent noises $\xi_t \sim \mathcal{N}(0, \sigma_{\mathbf{w}}^2 I_{d_1})$ and $\zeta_t \sim \mathcal{N}(0, \sigma_{\mathbf{v}}^2 I_{d_2})$
6:     $\mathbf{w}_{t+1} = \Pi_{\mathcal{W}}\Big(\mathbf{w}_t - \eta_{\mathbf{w},t}\big(\frac{1}{m}\sum_{j=1}^{m} \nabla_{\mathbf{w}} f(\mathbf{w}_t, \mathbf{v}_t; \mathbf{z}_{i_t^j}) + \xi_t\big)\Big)$
7:     $\mathbf{v}_{t+1} = \Pi_{\mathcal{V}}\Big(\mathbf{v}_t + \eta_{\mathbf{v},t}\big(\frac{1}{m}\sum_{j=1}^{m} \nabla_{\mathbf{v}} f(\mathbf{w}_t, \mathbf{v}_t; \mathbf{z}_{i_t^j}) + \zeta_t\big)\Big)$
8: **end for**
9: **Outputs:** $(\bar{\mathbf{w}}_T, \bar{\mathbf{v}}_T) = \frac{1}{T}\sum_{t=1}^{T}(\mathbf{w}_t, \mathbf{v}_t)$ or $(\mathbf{w}_T, \mathbf{v}_T)$

---

**Definition 3** (Differential Privacy). *A (randomized) algorithm $A$ is called $(\epsilon, \delta)$-differentially private (DP) if, for all neighboring datasets $S, S'$ and for all events $O$ in the output space of $A$, the following holds*

$$\mathbb{P}[A(S) \in O] \leq e^{\epsilon}\mathbb{P}[A(S') \in O] + \delta.$$

Our aim is to design a randomized algorithm satisfying $(\epsilon, \delta)$-DP which solves the empirical minimax problem:

$$\min_{\mathbf{w}\in\mathcal{W}} \max_{\mathbf{v}\in\mathcal{V}} \Big\{F_S(\mathbf{w}, \mathbf{v}) = \frac{1}{n}\sum_{i=1}^{n} f(\mathbf{w}, \mathbf{v}; \mathbf{z}_i)\Big\}. \quad (2)$$

Notice that in the standard ERM problem, which involves the minimization only with respect to $\mathbf{w}$, DP-SGD [Wu et al., 2017, Song et al., 2013, Bassily et al., 2019, Wang et al., 2020a] uses the gradient perturbation at each iteration. Specifically, at each iteration of this algorithm, a randomized gradient estimated from a random subset (mini-batch) of $S$ is perturbed by a Gaussian noise and then the model parameter is updated based on this noisy gradient.

Following the same spirit, DP-SGDA [Xie et al., 2018, Zhang et al., 2018] adds Gaussian noises per iteration to the randomized gradient mapping $(g_{\mathbf{w},t}, g_{\mathbf{v},t}) = (\frac{1}{m}\sum_{j=1}^{m} \nabla_{\mathbf{w}} f(\mathbf{w}_t, \mathbf{v}_t; \mathbf{z}_{i_t^j}), \frac{1}{m}\sum_{j=1}^{m} \nabla_{\mathbf{v}} f(\mathbf{w}_t, \mathbf{v}_t; \mathbf{z}_{i_t^j}))$ where the index of example $\mathbf{z}_{i_t^j}$ is from the mini-batch $I_t$. Then, the primal variable $\mathbf{w}$ is updated by gradient descent based on the noisy gradient $g_{\mathbf{w},t} + \xi_t$ and the dual variable $\mathbf{v}$ is updated by gradient ascent based on the noisy gradient $g_{\mathbf{v},t} + \zeta_t$. The pseudo-code for DP-SGDA is given in Algorithm 1. The noise levels $\sigma_{\mathbf{w}}, \sigma_{\mathbf{v}}$ are given

by (3) which will be specified soon in Section 3 in order to guarantee $(\epsilon, \delta)$-DP. The notations $\Pi_{\mathcal{W}}(\cdot)$ and $\Pi_{\mathcal{V}}(\cdot)$ denote the projections to $\mathcal{W}$ and $\mathcal{V}$, respectively. From now on, the notation $A$ denotes the DP-SGDA algorithm and its output is denoted by $A(S) = (A_{\mathbf{w}}(S), A_{\mathbf{v}}(S))$.

## 2.3 MEASURES OF UTILITY

Since the model $A(S)$ is only trained based on the training data $S$, its empirical behavior as measured by $F_S$ may not generalize well on test data. Our goal is to investigate the statistical behavior of $A(S)$ on the test data in terms of some population risk. However, unlike the standard statistical learning theory (SLT) setting where there is only a minimization of $\mathbf{w}$, we have different measures of population risk due to the minimax structure [Zhang et al., 2021a, Lei et al., 2021]. Let $\mathbb{E}[\cdot]$ denote the expectation with respect to the randomness of algorithm $A$ and data $S$. We are particularly interested in the following metrics.

**Definition 4** (Weak Primal-Dual (PD) Risk). *The weak primal-dual population risk of $A(S)$, denoted by $\triangle^w(A_{\mathbf{w}}(S), A_{\mathbf{v}}(S))$, is defined as*

$$\max_{\mathbf{v}\in\mathcal{V}} \mathbb{E}\big[F(A_{\mathbf{w}}(S), \mathbf{v})\big] - \min_{\mathbf{w}\in\mathcal{W}} \mathbb{E}\big[F(\mathbf{w}, A_{\mathbf{v}}(S))\big].$$

*The corresponding weak PD empirical risk, denoted by $\triangle_S^w(A_{\mathbf{w}}(S), A_{\mathbf{v}}(S))$, is defined as*

$$\max_{\mathbf{v}\in\mathcal{V}} \mathbb{E}\big[F_S(A_{\mathbf{w}}(S), \mathbf{v})\big] - \min_{\mathbf{w}\in\mathcal{W}} \mathbb{E}\big[F_S(\mathbf{w}, A_{\mathbf{v}}(S))\big].$$

**Definition 5** (Primal Risk). *The primal population risk of $A(S)$ is given by $R(A_{\mathbf{w}}(S)) = \max_{\mathbf{v}\in\mathcal{V}} F(A_{\mathbf{w}}(S), \mathbf{v})$ and empirical risk is defined by $R_S(A_{\mathbf{w}}(S)) = \max_{\mathbf{v}\in\mathcal{V}} F_S(A_{\mathbf{w}}(S), \mathbf{v})$, respectively. The excess primal population risk is defined as*

$$\mathbb{E}\big[R(A_{\mathbf{w}}(S)) - \min_{\mathbf{w}\in\mathcal{W}} R(\mathbf{w})\big].$$

*The corresponding excess primal empirical risk is then*

$$\mathbb{E}\big[R_S(A_{\mathbf{w}}(S)) - \min_{\mathbf{w}\in\mathcal{W}} R_S(\mathbf{w})\big].$$

Meanwhile, the strong PD risk defined as $\triangle^s(\mathbf{w}, \mathbf{v}) = \mathbb{E}\big[\sup_{\mathbf{v}'\in\mathcal{V}} F(\mathbf{w}, \mathbf{v}') - \inf_{\mathbf{w}'\in\mathcal{W}} F(\mathbf{w}', \mathbf{v})\big]$. We have $\triangle^w(A_{\mathbf{w}}(S), A_{\mathbf{v}}(S)) \leq \triangle^s(A_{\mathbf{w}}(S), A_{\mathbf{v}}(S))$ by applying Jensen's inequality. However, when $F$ is strongly-convex-strongly-concave, the point distance from the model $(A_{\mathbf{w}}(S), A_{\mathbf{v}}(S))$ to the true saddle point $(\mathbf{w}^*, \mathbf{v}^*) \in \arg\min_{\mathbf{w}\in\mathcal{W}} \max_{\mathbf{v}\in\mathcal{V}} F(\mathbf{w}, \mathbf{v})$ can be bounded by the weak PD population risk, i.e. $\mathbb{E}[\|A_{\mathbf{w}}(S) - \mathbf{w}^*\|_2^2 + \|A_{\mathbf{v}}(S) - \mathbf{v}^*\|_2^2] \leq \mathcal{O}(\triangle^w(A_{\mathbf{w}}(S), A_{\mathbf{v}}(S)))$. For certain problems, it is suffices to bound the weak PD risk, such as the learning problem for Markov decision process in Appendix A. The primal risk is more meaningful when one is concerned about the risk with respect to the primal variable, such as the AUC maximization problem.

# 3 MAIN RESULTS

In this section, we present our main theoretical results for DP-SGDA. For the privacy guarantee, we leverage the moments accountant method [Abadi et al., 2016], which implies tight privacy loss for adaptive Gaussian mechanisms with amplification by subsampling. Below we summarize a specific version of this method that suffices for our purpose.

**Theorem 1.** *Let (A1) hold true. Then, there exist constants $c_1, c_2$ and $c_3$ so that given the mini-batch size $m$ and total iterations $T$, for any $\epsilon < c_1 m^2 T/n^2$, Algorithm 1 is $(\epsilon, \delta)$-differentially private for any $\delta > 0$ if we choose*

$$\sigma_{\mathbf{w}} = \frac{c_2 G_{\mathbf{w}} \sqrt{T \log(1/\delta)}}{n\epsilon}, \ \sigma_{\mathbf{v}} = \frac{c_3 G_{\mathbf{v}} \sqrt{T \log(1/\delta)}}{n\epsilon}. \quad (3)$$

The proof of Theorem 1 is given in Appendix B.

**Remark 1.** *In practice, given privacy budget $\epsilon, \delta$ and parameters $m, T$, the constant $c_2$ and hence $\sigma$ can be found by grid search [Abadi et al., 2016]. Here we provide a set of parameters that satisfies the condition in that reference and our Theorem 1. That is, by choosing $\epsilon \leq 1, \delta \leq 1/n^2$ and $m = \max(1, n\sqrt{\epsilon/(4T)})$, then we have explicit values for the variances as $\sigma_{\mathbf{w}} = \frac{8 G_{\mathbf{w}} \sqrt{T \log(1/\delta)}}{n\epsilon}, \sigma_{\mathbf{v}} = \frac{8 G_{\mathbf{v}} \sqrt{T \log(1/\delta)}}{n\epsilon}$.*

**Remark 2.** *Our Algorithm 1 allows the application of independent noises $\xi_t, \zeta_t$ with different $\sigma_{\mathbf{w}}, \sigma_{\mathbf{v}}$, respectively. In Boob and Guzmán [2021], a uniform $\sigma$ is used (Theorem 5.4 or 7.4 there) for both primal and dual variables. In many examples, the primal and dual gradients $\nabla_{\mathbf{w}} f(\mathbf{w}_t, \mathbf{v}_t, \mathbf{z}_{i_t^j}), \nabla_{\mathbf{v}} f(\mathbf{w}_t, \mathbf{v}_t, \mathbf{z}_{i_t^j})$ enjoy different Lipschitz constants ($\ell_2$-sensitivity). Therefore, our treatment leads to a more delicate way of calibrating the variances of the Gaussian noises. As we shall see in the experiments in Section 4, this treatment enables Algorithm 1 to achieve better performance.*

In the subsequent subsections, we present our main contribution of this paper, i.e., the utility bounds of DP-SGDA for the convex-concave and nonconvex-strongly-concave cases, respectively.

## 3.1 CONVEX-CONCAVE CASE

In this subsection, we present the utility bound of DP-SGDA for the convex-concave case in terms of the weak PD risk of the output $(\bar{\mathbf{w}}_T, \bar{\mathbf{v}}_T)$ of Algorithm 1.

**Theorem 2.** *Assume the function $f$ is convex-concave. Assume $\mathcal{W}$ and $\mathcal{V}$ are bounded so that $\max_{\mathbf{w} \in \mathcal{W}} \|\mathbf{w}\|_2 \leq D_{\mathbf{w}}$, $\max_{\mathbf{v} \in \mathcal{V}} \|\mathbf{v}\|_2 \leq D_{\mathbf{v}}$. And let $D = \max\{D_{\mathbf{w}}, D_{\mathbf{v}}\}$. Let the stepsizes $\eta_{\mathbf{w},t} = \eta_{\mathbf{v},t} = \eta$ for all $t \in [T]$ with some $\eta > 0$. Under one of the condition*

*a) Assumption (A1) and (A3) hold true and we choose $T \asymp n$ and $\eta \asymp 1/\left(\max\{\sqrt{n}, \sqrt{d \log(1/\delta)}/\epsilon\}\right)$,*

*b) or Assumption (A1) holds true and we choose $T \asymp n^2$ and $\eta \asymp 1/\left(n \max\{\sqrt{n}, \sqrt{d \log(1/\delta)}/\epsilon\}\right)$,*

*then Algorithm 1 satisfies*

$$\triangle^w(\bar{\mathbf{w}}_T, \bar{\mathbf{v}}_T) = \mathcal{O}\left(\max\left\{\frac{1}{\sqrt{n}}, \frac{\sqrt{d \log(1/\delta)}}{n\epsilon}\right\}\right).$$

Its detailed proof can be found in Appendix C. The proof mainly relies on the concept of stability [Bousquet and Elisseeff, 2002, Charles and Papailiopoulos, 2018, Hardt et al., 2016, Kuzborskij and Lampert, 2018]. Specifically, the weak PD population risk can be decomposed as follows:

$$\triangle^w(\bar{\mathbf{w}}_T, \bar{\mathbf{v}}_T) = \triangle^w(\bar{\mathbf{w}}_T, \bar{\mathbf{v}}_T) - \triangle_S^w(\bar{\mathbf{w}}_T, \bar{\mathbf{v}}_T) \\ + \triangle_S^w(\bar{\mathbf{w}}_T, \bar{\mathbf{v}}_T), \quad (4)$$

where the term $\triangle^w(\bar{\mathbf{w}}_T, \bar{\mathbf{v}}_T) - \triangle_S^w(\bar{\mathbf{w}}_T, \bar{\mathbf{v}}_T)$ is the generalization error and $\triangle_S^w(\bar{\mathbf{w}}_T, \bar{\mathbf{v}}_T)$ is the optimization error.

The estimation for the optimization error can be conducted by standard techniques [Nemirovski et al., 2009]. We give a self-contained proof in Appendix C.1. The generalization error is estimated using a concept of weak stability [Lei et al., 2021]. Specifically, we say the randomized algorithm $A$ is $\varepsilon$-weakly-stable if, for any neighboring sets $S, S'$ differing at one single datum, there holds

$$\sup_{\mathbf{z}} \left(\sup_{\mathbf{v} \in \mathcal{V}} \mathbb{E}_A[f(A_{\mathbf{w}}(S), \mathbf{v}; \mathbf{z}) - f(A_{\mathbf{w}}(S'), \mathbf{v}; \mathbf{z})] \right. \\ \left. + \sup_{\mathbf{w} \in \mathcal{W}} \mathbb{E}_A[f(\mathbf{w}, A_{\mathbf{v}}(S); \mathbf{z}) - f(\mathbf{w}, A_{\mathbf{v}}(S'); \mathbf{z})]\right) \leq \varepsilon.$$

We know from Lei et al. [2021] that $\varepsilon$-weak-stability implies $\triangle^w(A_{\mathbf{w}}(S), A_{\mathbf{v}}(S)) - \triangle_S^w(A_{\mathbf{w}}(S), A_{\mathbf{v}}(S)) \leq \varepsilon$.

In Appendix C.2, we prove the weak stability of DP-SGDA (i.e. Algorithm 1) for both smooth and nonsmooth cases. Putting the estimations for the optimization error and generalization error into (4) can yield the bound in Theorem 2. We end this subsection with some remarks.

**Remark 3.** *The utility bound $\mathcal{O}\left(\max\left\{\frac{1}{\sqrt{n}}, \frac{\sqrt{d \log(1/\delta)}}{n\epsilon}\right\}\right)$ is optimal for convex-concave minimax problem. A lower bound with the same order has been established in the convex ERM setting [Bassily et al., 2014, 2019, Feldman et al., 2020] and the measure of utility is given by $\mathbb{E}[F(A_{\mathbf{w}}(S)) - \min_{\mathbf{w} \in \mathcal{W}} F(\mathbf{w})]$. Here we slightly abuse the notation to indicate $F$ as the population risk and $A_{\mathbf{w}}(S)$ as the algorithm for the ERM problem. Since the convex-concave minimax problem is a special case of convex ERM problems when the dual variable is constant, this lower bound also applies to our setting.*

**Remark 4.** *The same optimal utility was claimed in Boob and Guzmán [2021]. Yet our results also possess two theoretical gains compared to theirs. Firstly, when the smoothness assumption holds, Part a) in our Theorem 2 shows the optimal utility with $T = \mathcal{O}(n)$ iterations and $\mathcal{O}(n^{3/2})$ gradient*

*computations by Remark 1, while their single-looped algorithm (Algorithm 1 there) requires $\mathcal{O}(n^2)$ gradient computations in their Theorem 5.4. They further improved the gradient complexity to $\mathcal{O}(n^{3/2}\log(n))$ in Theorem 7.4, which, however, requires an extra subroutine algorithm (inner-loop) (Algorithm 2 there). Secondly, we also derive the same optimal bound with only Lipschitz continuous assumption for the nonsmooth case which was not addressed in Boob and Guzmán [2021].*

## 3.2 NONCONVEX-STRONGLY-CONCAVE CASE

We proceed to the case when $f$ is non-convex-strongly-concave. In this case, we can present utility bounds of DP-SGDA in terms of the primal excess risk, i.e., $R(\mathbf{w}_T) - \min_{\mathbf{w}\in\mathcal{W}} R(\mathbf{w})$, where $\mathbf{w}_T$ is the last iterate of Algorithm 1. Generally speaking, a saddle point may not always exist without the convexity assumption. Since our goal in this paper is to find global optima, we assume that the saddle point of the empirical minimax problem exists, i.e., there exists $(\hat{\mathbf{w}}_S, \hat{\mathbf{v}}_S)$ such that, for any $\mathbf{w}\in\mathcal{W}$ and $\mathbf{v}\in\mathcal{V}$,

$$F_S(\hat{\mathbf{w}}_S, \mathbf{v}) \le F_S(\hat{\mathbf{w}}_S, \hat{\mathbf{v}}_S) \le F_S(\mathbf{w}, \hat{\mathbf{v}}_S).$$

To estimate the primal excess risk, we define $R_S^* = \min_{\mathbf{w}\in\mathcal{W}} R_S(\mathbf{w})$, and $R^* = \min_{\mathbf{w}\in\mathcal{W}} R(\mathbf{w})$. Then, for any $\mathbf{w}^* \in \arg\min_{\mathbf{w}} R(\mathbf{w})$ we have the error decomposition:

$$\begin{aligned}
\mathbb{E}[R(\mathbf{w}_T) - R^*] &= \mathbb{E}[R(\mathbf{w}_T) - R_S(\mathbf{w}_T)] + \mathbb{E}[R_S(\mathbf{w}_T) - R_S^*] \\
&\quad + \mathbb{E}[R_S^* - R_S(\mathbf{w}^*)] + \mathbb{E}[R_S(\mathbf{w}^*) - R(\mathbf{w}^*)] \\
&\le \mathbb{E}[R(\mathbf{w}_T) - R_S(\mathbf{w}_T)] + \mathbb{E}[R_S(\mathbf{w}^*) - R(\mathbf{w}^*)] \\
&\quad + \mathbb{E}[R_S(\mathbf{w}_T) - R_S^*], \quad\quad\quad (5)
\end{aligned}$$

where the last inequality follows from the fact that $R_S^* - R_S(\mathbf{w}^*) \le 0$ since $R_S^* = \min_{\mathbf{w}\in\mathcal{W}} R_S(\mathbf{w})$. The term $\mathbb{E}[R_S(\mathbf{w}_T) - R_S^*]$ is the *optimization error* which characterizes the discrepancy between the primal empirical risk of an output of Algorithm 1 and the least possible one. The term $\mathbb{E}[R(\mathbf{w}_T) - R_S(\mathbf{w}_T)] + \mathbb{E}[R_S(\mathbf{w}^*) - R(\mathbf{w}^*)]$ is called *generalization error* which measures the discrepancy between the primal population risk and the empirical one. The estimations for these two errors are described as follows.

**Optimization Error.** The next theorem characterizes the primal empirical risk of DP-SGDA under the PL-SC assumption.

**Theorem 3.** *Assume Assumptions (A1) and (A2) hold true, and the function $F_S(\mathbf{w}, \cdot)$ is $\rho$-strongly concave and $F_S(\cdot, \mathbf{v})$ satisfies $\mu$-PL condition. Assume $\mathcal{V}$ is bounded. Let $\kappa = L/\rho$. If we choose $\eta_{\mathbf{w},t} \asymp \frac{1}{\mu t}$ and $\eta_{\mathbf{v},t} \asymp \frac{\kappa^{2.5}}{\mu^{1.5}t^{2/3}}$, then*

$$\mathbb{E}[R_S(\mathbf{w}_{T+1}) - R_S^*] = \mathcal{O}\Big(\frac{\kappa^{3.5}}{\mu^{2.5}}\Big(\frac{1/m + d(\sigma_{\mathbf{w}}^2 + \sigma_{\mathbf{v}}^2)}{T^{2/3}}\Big)\Big).$$

We provide the proof of Theorem 3 in Appendix D.1. In the non-private setting, i.e. $\sigma_{\mathbf{w}} = \sigma_{\mathbf{v}} = 0$, Theorem 3 implies that the convergence rate in terms of the primal empirical risk is of the order $\mathcal{O}(\frac{\kappa^{3.5}}{\mu^{2.5}T^{2/3}})$, which is a new result even in the non-private case as far as we are aware of.

In Lin et al. [2020], the local convergence of SGDA in the non-private case was proved in terms of the metric $\mathbb{E}_\tau[\|\nabla R_S(\mathbf{w}_\tau)\|_2^2]$ where $\tau$ is chosen uniformly at random from the set $\{1, 2, \dots, T\}$. Our analysis is much more involved since it proves the global convergence of the last iterate $\mathbf{w}_T$. Our main idea is to prove the coupled recursive inequalities for two terms, i.e., $a_t = R_S(\mathbf{w}_t) - R_S^*$ and $b_t = \|\mathbf{v}_t - \hat{\mathbf{v}}_S(\mathbf{w}_t)\|_2^2$ where $\hat{\mathbf{v}}_S(\mathbf{w}_t) = \arg\max_{\mathbf{v}\in\mathcal{V}} F_S(\mathbf{w}_t, \mathbf{v})$, and then carefully derive the the convergence rate for $a_t + \lambda_t b_t$ by choosing $\lambda_t$ appropriately. The convergence rate and its proof can be of interest in their own right. One can find more detailed arguments in Appendix D.1.

**Generalization Error.** We present the bound for the generalization error which is proved again using the stability approach.

We begin with a discussion of the saddle points. While the saddle point $(\hat{\mathbf{w}}_S, \hat{\mathbf{v}}_S)$ may not be unique, $\hat{\mathbf{v}}_S$ must be unique if $F_S(\mathbf{w}, \mathbf{v})$ is strongly-concave in $\mathbf{v}$ (see Proposition 1 in Appendix D). Therefore, we can define $\pi_S(\mathbf{w})$ the projection of $\mathbf{w}$ to the set of saddle points, as $\Omega_S = \{\hat{\mathbf{w}}_S : (\hat{\mathbf{w}}_S, \hat{\mathbf{v}}_S) \in \arg\min_{\mathbf{w}\in\mathcal{W}} \max_{\mathbf{v}\in\mathcal{V}} F_S(\mathbf{w}, \mathbf{v})\} = \{\hat{\mathbf{w}}_S : \hat{\mathbf{w}}_S \in \arg\min_{\mathbf{w}\in\mathcal{W}} F_S(\mathbf{w}, \hat{\mathbf{v}}_S)\}$.

Recall that $\mathbf{w}_T$ is the iterate of DP-SGDA at time $T$ based on the training data $S$. Likewise, we denote by $\mathbf{w}_T'$ based on the training set $S'$ which differs from $S$ at one single datum. Due to the possibly multiple saddle points, we need the following critical assumption for estimating the generalization error.

**Assumption 4 (A4).** *For the (randomized) algorithm DP-SGDA, assume that $\pi_{S'}(\pi_S(\mathbf{w}_T)) = \pi_{S'}(\mathbf{w}_T')$ for any neighboring sets $S$ and $S'$.*

Assumption **(A4)** was introduced in Charles and Papailiopoulos [2018] for studying the stability of SGD in the non-convex case which only involves the minimization over $\mathbf{w}$. In our case, **(A4)** holds true whether the saddle point is unique (e.g., $F_S$ is strongly-convex and strongly-concave) or the two sets of saddle points based on $S$ and $S'$, i.e. $\Omega_S$ and $\Omega_{S'}$ do not change too much. Since our algorithm satisfies $(\epsilon, \delta)$-DP it means that the distributions of $\mathbf{w}_T$ and $\mathbf{w}_T'$ generated from two neighboring sets $S$ and $S'$ are "close", which indicates $\sup_{S,S'} \|\pi_{S'}(\pi_S(\mathbf{w}_T)) - \pi_{S'}(\mathbf{w}_T')\|_2$ can be small. Proving such statement serves as an interesting open problem.

Now we can state the results on the generalization error.

**Theorem 4 (Generalization Error).** *Assume Assumptions (A1), (A3) and (A4) hold true, and assume the function $f(\mathbf{w}, \cdot; \mathbf{z})$ is $\rho$-strongly concave and $F_S(\cdot, \mathbf{v})$ satisfies $\mu$-PL*

*condition. Let $\kappa = L/\rho$. If $\mathbb{E}[R_S(\mathbf{w}_T) - R_S^*] \leq \varepsilon_T$, then*

$$\mathbb{E}[R(\mathbf{w}_T) - R_S(\mathbf{w}_T)] \leq (1+\kappa)G_{\mathbf{w}}\Big(\sqrt{\frac{\varepsilon_T}{2\mu}} + \frac{1}{n}\sqrt{\frac{G_{\mathbf{w}}^2}{4\mu^2} + \frac{G_{\mathbf{v}}^2}{\rho\mu}}\Big),$$

*and*

$$\mathbb{E}[R_S(\mathbf{w}^*) - R(\mathbf{w}^*)] \leq \frac{4G_{\mathbf{v}}^2}{\rho n}.$$

The proof of Theorem 4 is provided in Appendix D.2.

**Remark 5.** *The generalization error bounds given in Theorem 4 indicate that if the optimization error $\mathbb{E}[R_S(\mathbf{w}_T) - R_S^*]$ is small then the generalization error will be small. This is consistent with the observation in the stability and generalization analysis of SGD [Charles and Papailiopoulos, 2018, Hardt et al., 2016, Lei and Ying, 2021] for the minimization problems in the sense of "optimization can help generalization".*

We can derive the following utility bound for DP-SGDA by combining the results in Theorems 4 and 3.

**Theorem 5.** *Under the same assumptions of Theorem 4, if we choose $T \asymp n$, $\eta_{\mathbf{w},t} \asymp \frac{1}{\mu t}$ and $\eta_{\mathbf{v},t} \asymp \frac{\kappa^{2.5}}{\mu^{1.5}t^{2/3}}$, then*

$$\mathbb{E}[R(\mathbf{w}_{T+1}) - R^*] = \mathcal{O}\Big(\frac{\kappa^{2.75}}{\mu^{1.75}}\Big(\frac{1}{n^{1/3}} + \frac{\sqrt{d\log(1/\delta)}}{n^{5/6}\epsilon}\Big)\Big).$$

The proof can be found in Appendix D.3.

## 4  EXPERIMENTS

In this section, we evaluate the performance of DP-SGDA by taking AUC maximization as an example. Due to space limitation, we present the most significant information and results of our experiments while more detailed information and additional results are given in Appendix E and F.

### 4.1  EXPERIMENTAL SETTINGS

**Baseline Model.** We perform experiments on the problem of AUC maximization with the least square loss to evaluate the DP-SGDA algorithm in linear and non-linear settings (two-layer multilayer perceptron (MLP)). In this case, AUC maximization can be formulated as

$$\min_{\theta\in\Theta} \mathbb{E}_{\mathbf{z},\mathbf{z}'}[(1 - h(\theta;\mathbf{x}) + h(\theta;\mathbf{x}'))^2 | y = 1, y' = -1],$$

where $h : \Theta \times \mathbb{R}^d \to \mathbb{R}$ is the scoring function. As shown in Ying et al. [2016], it is equivalent to a minimax problem:

$$\min_{\mathbf{w}=(\theta,a,b)} \max_{\mathbf{v}} \mathbb{E}_{\mathbf{z}}[f(\theta, a, b, \mathbf{v}; \mathbf{z})],$$

where $f = (1 - p)(h(\theta;\mathbf{x}) - a)^2\mathbb{I}[y = 1] + p(h(\theta;\mathbf{x}) - b)^2\mathbb{I}[y = -1] + 2(1 + \mathbf{v})(ph(\theta;\mathbf{x})\mathbb{I}[y = -1] - (1 - p)h(\theta;\mathbf{x})\mathbb{I}[y = 1])) - p(1 - p)\mathbf{v}^2$ and $p = \mathbb{P}[y = 1]$.

When $h$ is a linear function, the AUC learning objective above is convex-strongly-concave. On the other hand, when $h$ is a MLP function, it becomes a nonconvex-strongly-concave minimax problem. In addition, following Liu et al. [2020], we use Leaky ReLU as an activation function for MLP. It was shown in their paper the empirical AUC objective satisfies the PL condition with this choice of $h$. Without a special statement, we set 256 as the number of hidden units in MLP and 64 as the mini-batch size during the training.

**Datasets and Evaluation Metrics.** Our experiments are based on three popular datasets, namely ijcnn1 [Chang and Lin, 2011], MNIST [LeCun et al., 1998], and Fashion-MNIST [Xiao et al., 2017] that have been used in previous studies. For MNIST and Fashion-MNIST, following Gao et al. [2013], Ying et al. [2016], we transform their classes into binary classes by randomly partitioning the data into two groups, each with an equal number of classes. For ijcnn1, we randomly split its original training set into new training (80%) and testing (20%) sets. For MNIST and Fashion-MNIST, we use their original training set and testing set. For each method, the reported performance is obtained by averaging the AUC scores on the test set according to 5 random seeds (for initial $\mathbf{w}$ and $\mathbf{v}$, sampling and noise generation).

**Privacy Budget Settings.** In the experiments, we set up five privacy levels from small to large: $\epsilon \in \{0.1, 0.5, 1, 5, 10\}$. We also consider three different $\delta$ from $\{1e-4, 1e-5, 1e-6\}$. Due to space limitation, we only report the performance when $\delta = 1e-6$. More results can be found in Appendix F. To estimate the Lipschitz constants $G_{\mathbf{w}}$ and $G_{\mathbf{v}}$ (in Theorem 1), we first run the algorithms without adding noise. Then we calculate the maximum gradient norms of AUC loss w.r.t $\mathbf{w}$ and $\mathbf{v}$ and assign them as $G_{\mathbf{w}}$ and $G_{\mathbf{v}}$, respectively. According to these parameters, we calculate the noise parameter $\sigma$ by applying autodp[1], which is widely used in the existing works [Wang et al., 2019b].

**Compared Algorithms.** Boob and Guzmán [2021] is the only existing paper that considers differential privacy in the convex-concave minimax problem. Therefore, we use their single-loop NSEG algorithm as our baseline method on the AUC optimization under the linear setting.

### 4.2  RESULTS

We report our evaluation and results on the utility and privacy trade-off of the DP-SGDA. Then we follow the experiment design by [Abadi et al., 2016] to study the effect of the parameters - hidden units and batch sizes.

---

[1] https://github.com/yuxiangw/autodp

| Dataset | ijcnn1 | | | MNIST | | | Fashion-MNIST | | |
|---|---|---|---|---|---|---|---|---|---|
| Algorithm | Linear | | MLP | Linear | | MLP | Linear | | MLP |
| | NSEG | DP-SGDA | DP-SGDA | NSEG | DP-SGDA | DP-SGDA | NSEG | DP-SGDA | DP-SGDA |
| Original | 92.191 | 92.448 | 96.609 | 93.306 | 93.349 | 99.546 | 96.552 | 96.523 | 98.020 |
| $\epsilon$=0.1 | 90.106 | 91.110 | 92.763 | 91.247 | 91.858 | 97.878 | 95.446 | 95.468 | 95.692 |
| $\epsilon$=0.5 | 90.346 | 91.357 | 95.840 | 91.324 | 92.058 | 98.656 | 95.530 | 95.816 | 96.988 |
| $\epsilon$=1 | 90.355 | 91.371 | 96.167 | 91.330 | 92.070 | 98.705 | 95.534 | 95.834 | 97.102 |
| $\epsilon$=5 | 90.363 | 91.383 | 96.294 | 91.334 | 92.078 | 98.742 | 95.538 | 95.848 | 97.198 |
| $\epsilon$=10 | 90.363 | 91.386 | 96.297 | 91.334 | 92.080 | 98.747 | 95.539 | 95.850 | 97.213 |

Table 2: *Comparison of AUC performance in NSEG and DP-SGDA (Linear and MLP settings) on three datasets with different $\epsilon$ and $\delta$=1e-6. The "Original" means no noise ($\epsilon = \infty$) is added in the algorithms.*

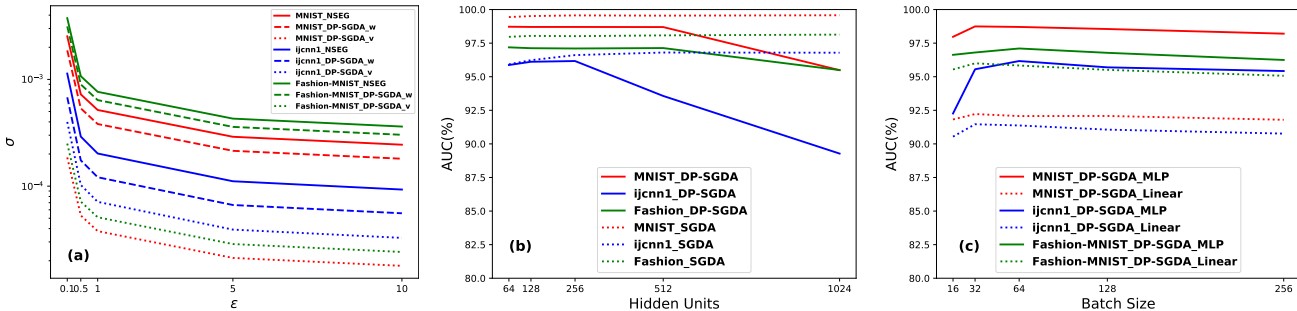

Figure 1: *(a) Comparison of $\sigma$ for NSEG and DP-SGDA (Linear setting) on three datasets with different $\epsilon$ and $\delta$=1e-6. (b)Comparison of AUC performance for SGDA and DP-SGDA in MLP settings on three datasets with different hidden units and $\epsilon$=1 and $\delta$=1e-6. (c) Comparison of AUC performance for DP-SGDA (Linear and MLP settings) on three datasets with different batch size and $\epsilon$=1 and $\delta$=1e-6.*

**General AUC Performance vs Privacy.** The general performance of all algorithms under linear and MLP settings of AUC optimization is shown in Table 2. Since the standard deviation of the AUC performance is around $[0, 0.1\%]$ and the difference between different algortihms is very small, we only report the average AUC performance. First, without adding noise into gradients, we can find the NSEG method and our DP-SGDA method have similar performance under the linear case. Furthermore, we can find the performance of the DP-SGDA with MLP model can outperform linear models on all datasets. This is because non-linear models have better expression power and therefore it can learn more information among features than linear models. Second, by adding noise into the gradients, we can find the AUC performance of all models is decreased on all datasets. However, by increasing the privacy budget $\epsilon$, the AUC performance is increased. The reason is that $\epsilon$ and $\sigma$ have opposite trends according to equation (3). The relation between $\epsilon$ and AUC score also verifies our Theorem 2 and Theorem 5. Third, to verify our statement in Remark 2, we compare the $\sigma$ values from NSEG and DP-SGDA on all datasets in Figure 1(a). From the figure, it is clear that the $\sigma$ from NSEG is larger than ours in all $\epsilon$ settings since it is calibrated based on the gradients' sensitivity from both $\mathbf{w}$ and $\mathbf{v}$. In fact, the sensitivity w.r.t. $\mathbf{v}$ is small as it is a one-dimensional variable for AUC maximization. Therefore, NSEG leads to overestimate

on the noise addition towards $\mathbf{v}$. From Table 2 we observe our DP-SGDA achieves better AUC score than NSEG under the same privacy budget.

**Different Hidden Units.** In DP-SGDA under the MLP setting, the hidden unit is one of the most important factors affecting the model performance. Therefore, we compare the AUC performance with respect to the different hidden units in Figure 1(b). If we provide a small number of hidden units, the model will suffer from poor generalization capability. Using a large number of hidden units will make the model easier to fit the training set. For SGDA (non-private) training, it is often helpful to apply a large number of hidden units, as long as the model does not overfit. In agreement with this intuition, we find the model performance improves with increasing hidden units in Figure 1(b). However, for DP-SGDA training, more hidden units increase the sensitivity of the gradients, which leads to more noise added at each update. Therefore, in contrast to the non-private setting, we find the AUC performance decreases when the number of hidden units increases.

**Different Mini-Batch Size.** From Theorem 1 and Theorem 3, we find mini-batch size can influence the Gaussian noise variances $\sigma_{\mathbf{w}}^2$ and $\sigma_{\mathbf{v}}^2$ as well as the convergence rate. Selecting the mini-batch size must balance two conflicting objectives. On one hand, a small mini-batch size may lead to

sub-optimal performance. On the other hand, for large batch sizes, the added noise has a smaller relative effect. Therefore, we show the AUC score for DP-SGDA with different mini-batch sizes in Figure 1(c). The experimental results show that the mini-batch size has a relatively large impact on the AUC performance when the mini-batch size is small.

# 5 CONCLUSION

In this paper, we have used algorithmic stability to conduct utility analysis of the DP-SGDA algorithm for minimax problems under DP constraints. For the convex-concave setting, we proved that DP-SGDA can attain an optimal rate $\mathcal{O}\left(\frac{1}{\sqrt{n}} + \frac{\sqrt{d \log(1/\delta)}}{n\epsilon}\right)$ in terms of the weak primal-dual population risk while providing $(\epsilon, \delta)$-DP for both smooth and nonsmooth cases. For the nonconvex-strongly-concave case, assuming that the empirical risk satisfies the PL condition we proved the excess primal population risk of DP-SGDA can achieve a utility bound $\mathcal{O}\left(\frac{1}{n^{1/3}} + \frac{\sqrt{d \log(1/\delta)}}{n^{5/6}\epsilon}\right)$. Experiments on three benchmark datasets illustrate the effectiveness of DP-SGDA.

For future work, it would be interesting to improve the utility bound for the nonconvex-strongly-convex setting. It also remains unclear to us how to establish the utility bound for DP-SGDA when gradient clipping techniques are enforced at each iteration. Finally, it would also be interesting to evaluate the performance of DP-SGDA on other motivating examples such as GAN, MDP and robust optimization.

### Acknowledgements

The work is supported by SUNY-IBM AI Alliance Research and NSF grants (IIS-1816227, IIS-2008532, IIS-2103450, IIS-2110546 and DMS-2110836). The authors would also like to thank Dr. Guzmán and Dr. Boob for helpful discussions on differential privacy for minimax problems and for pointing out a gap in the proof of Lemma 3 in the Appendix in an earlier version of the paper.

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
