# OpenReview forum: "Differentially Private SGDA for Minimax Problems"
_auai.org/UAI/2022/Conference — UAI 2022 Poster_

### Official Review · Reviewer_FQfx · 2022-03-24

**Q2(1) Originality/Novelty:** 3
**Q2(2) Significance/Impact:** 3
**Q2(3) Correctness/Technical Quality:** 3
**Q2(6) Clarity Of Writing:** 3
**Q6 Overall Score:** 5
**Q8 Confidence In Your Score:** 2

**Q1 Summary And Contributions:**

This work studies the well-studied method SGDA with the considerations of differential privacy. The main contributions are that 1) in the convex-concave setting, this work gives the optimal utility rate in terms of weak primal-dual population risk in both smooth and non-smooth cases; 2) in the nonconvex-strongly-concave case, this work gives the first result in terms of primal population risk.

**Q2 Assessment Of The Paper:**

More detailed information regarding each of these aspects is given below:

**Q2(4) Quality Of Experiments (Optional):**

2: Fair: The experimental evaluation is weak: important baselines are missing, or the results do not adequately support the main claims.

**Q2(5) Reproducibility:**

3: Good: Key resources (e.g., proofs, code, data) are available and key details (e.g., proofs, experimental setup) are sufficiently well-described for competent researchers to confidently reproduce the main results.

**Q3 Main Strengths:**

1. The problem that this work investigates is well-motivated and studied since 1) SGDA models a large body of optimization problems such as AUC maximization and GAN; 2) DP considers data privacy which is commonly required in practical applications.
2. The theoretical contribution is sufficient: 1) the same result in the case of C-C, Lip, S, but with better computational efficiency; 2) the first result in C-C, Lip case without smoothness assumption; 3) the first result in PL-SC, Lip, S case.
3. The overall paper is well-written.

**Q4 Main Weakness:**

My only concern is about the experiment part, where most experiments seem not that meaningful:
1) In Table 2, MLP models are better than linear models for sure. The comparison between linear and non-linear models is not meaningful, in my opinion.
2) In Figure 1(c), I do not quite get the message from the experiment of this part. Can this result be served as a guide of the choice of the batch size?

**Q5 Detailed Comments To The Authors:**

See Q3 and Q4.

**Q7 Justification For Your Score:**

Mainly see Q3.

**Q9 Complying With Reviewing Instructions:**

1: Yes.

---

### Official Review · Reviewer_oJqE · 2022-04-12

**Q2(1) Originality/Novelty:** 3
**Q2(2) Significance/Impact:** 3
**Q2(3) Correctness/Technical Quality:** 3
**Q2(6) Clarity Of Writing:** 4
**Q6 Overall Score:** 7
**Q8 Confidence In Your Score:** 2

**Q1 Summary And Contributions:**

The paper studies convergence and generalization properties of a differentially private SGDA algorithm for finding saddle points in min-max problems.
An optimal convergence rate is proven for convex-concave problems, and a guarantee on convergence rate is also given for non-convex-strongly-concave problems (under an assumption on the risk that has been used in prior work differentially private SGD). Finally, experiments are performed to study the behavior of these losses in AUC maximization.

**Q2 Assessment Of The Paper:**

More detailed information regarding each of these aspects is given below:

**Q2(4) Quality Of Experiments (Optional):**

3: Good: The experimental evaluation is adequate, and the results convincingly support the main claims.

**Q2(5) Reproducibility:**

4: Excellent: Key resources (e.g., proofs, code, data) are available and key details (e.g., proof sketches, experimental setup) are comprehensively described for competent researchers to confidently and easily reproduce the main results.

**Q3 Main Strengths:**

To the best of my knowledge, the paper provides novel results on a rather difficult problem.
Differential privacy and min-max optimization are important topics, hence the results may be of interest to people working on the theory of both topics.
Writing is clear, even though the paper has many technical details.

**Q4 Main Weakness:**

Since my knowledge regarding analyses of SGD under differential privacy constraints is limited, I might be missing some important related work and weaknesses. Yet no obvious weaknesses come to mind after reading the paper.

**Q5 Detailed Comments To The Authors:**

Some experiments in settings other than AUC maximization may have been nice, though the scope of existing experiments is not too bad.

**Q7 Justification For Your Score:**

As far as I can see the paper provides a solid and novel analysis of a difficult problem that is relevant to many applications, without obvious flaws at least to my eye.

**Q9 Complying With Reviewing Instructions:**

1: Yes.

---

### Official Review · Reviewer_ZEPG · 2022-04-13

**Q2(1) Originality/Novelty:** 3
**Q2(2) Significance/Impact:** 2
**Q2(3) Correctness/Technical Quality:** 4
**Q2(6) Clarity Of Writing:** 4
**Q6 Overall Score:** 7
**Q8 Confidence In Your Score:** 3

**Q1 Summary And Contributions:**

This theory paper extends the theoretical analaysis of DP-SGD carried out by  [ Bassily et al., 2019, 2020, Feldman et al., 2020, Song et al., 2013, Wang et al., 2021a, 2020a, 2019b, Wu et al., 2017, Zhou et al., 2020] to the minimax problem where there is DP descent to one part of the model and DP ascent to the other part. There are two cases studied: Convex-concave and non-convex-strongly-concave (convexity / concavity referring to the properties of the two parts of the loss function).

**Q2 Assessment Of The Paper:**

More detailed information regarding each of these aspects is given below:

**Q2(4) Quality Of Experiments (Optional):**

3: Good: The experimental evaluation is adequate, and the results convincingly support the main claims.

**Q2(5) Reproducibility:**

3: Good: Key resources (e.g., proofs, code, data) are available and key details (e.g., proofs, experimental setup) are sufficiently well-described for competent researchers to confidently reproduce the main results.

**Q3 Main Strengths:**

- This is an important problem: the convex problems are studied a lot in DP-SGD literature, and especially focusing now on algorithms that provide only guarantees for the final model. This would be natural and desirable also for training of GANs, as having some convex algorithms that lead to reasonable guarantees to the final model would potentially improve the quality of DP synthetic data a lot.

-  Very well written, seems to be able to capture the essentials of DP-SGD results in this setting.



**Q4 Main Weakness:**


I feel there is at least some literature missing. Relevant in this setting, I think, would be the 'privacy amplification by iteration' - line of research starting from

Feldman, Vitaly, et al. "Privacy amplification by iteration." 2018 IEEE 59th Annual Symposium on Foundations of Computer Science (FOCS). IEEE, 2018.

and leading, with many steps, to

Chourasia, Rishav, Jiayuan Ye, and Reza Shokri. "Differential Privacy Dynamics of Langevin Diffusion and Noisy Gradient Descent." Advances in Neural Information Processing Systems 34 (2021).


There is no clear connection given between GANs and the problem setting considered, I don't immediately see if GAN training falls into these categories.


**Q5 Detailed Comments To The Authors:**

Could you comment on the 'Privacy amplification by iteration' - research: could you possible obtain privacy guarantees that hold only for the final model, in a 'convex-concave' or 'strongly convex-strongly concave' situation, for example?

Does the GAN problem mentioned in Section 1.1 fall in to these settings (Convex-concave or non-convex-strongly-concave) ?

**Q7 Justification For Your Score:**

Very well written theory paper on an important problem.

**Q9 Complying With Reviewing Instructions:**

1: Yes.

---

### Official Review · Reviewer_3vGX · 2022-04-14

**Q2(1) Originality/Novelty:** 2
**Q2(2) Significance/Impact:** 2
**Q2(3) Correctness/Technical Quality:** 3
**Q2(6) Clarity Of Writing:** 3
**Q6 Overall Score:** 6
**Q8 Confidence In Your Score:** 4

**Q1 Summary And Contributions:**

This paper uses the algorithmic stability approach to establish the utility of DP-SGDA in different settings.  Particularly, the authors prove that the proposed algorithm (DP-SGDA) can achieve an optimal utility rate in terms of the weak primal-dual population risk in both smooth and non-smooth cases. Further, the authors provide the utility analysis in the nonconvex-strongly-concave setting, and the convergence analysis in nonconvex setting.

**Q2 Assessment Of The Paper:**

More detailed information regarding each of these aspects is given below:

**Q2(4) Quality Of Experiments (Optional):**

2: Fair: The experimental evaluation is weak: important baselines are missing, or the results do not adequately support the main claims.

**Q2(5) Reproducibility:**

3: Good: Key resources (e.g., proofs, code, data) are available and key details (e.g., proofs, experimental setup) are sufficiently well-described for competent researchers to confidently reproduce the main results.

**Q3 Main Strengths:**

1)	Theoretical analyses are solid, including the convergence analysis and  utility analysis.

2)	Consider both the non-smooth and DP case.


**Q4 Main Weakness:**

1)	An overclaimed statement. In the Abstract or the summarized contributions, the authors claim that ``DP-SGDA’’ achieve the optimal rate.  While, I can not find any solid results to support this.

2)	The experimental section is weak.


**Q5 Detailed Comments To The Authors:**

This paper considers an interesting problem and provides some solid theoretical results. However, there are following problems

1) An overclaimed statement. In the Abstract or the summarized contributions, the authors claim that ``DP-SGDA’’ achieve the optimal rate.  If the ``optimal’’ means achieving the lower bound of such type algorithms?  If so, it’s better to provide the corresponding theoretical bond.

2) The statement ``Our analysis is much more involved since it proves the global convergence of the last
iterate $w-T$. ’’. This statement seems not true.  Because, if one choose to analysis the gap of  objective function value, then  one can always obtain the global convergence of the  last iterate $w-T$; and if one choose  to analysing the norm of gradient one can  obtain the global convergence of the excepted gradient norm.

3) The experiments are not sufficient to support the theoretical results.  As shown in Table 1, the proposed method has a better complexity than NSEG, while this is not verified in the experiments.

4) In section 4.2, the authors claim that `` we can find the NSEG method and our DP-SGDA method have similar performance under the linear case.’’. In fact, NSEG is worse than DP-SGDA (even without adding noise). Is such result consistent to the theoretical results?

5) There is an experiment of the ``Different Hidden Units’’. What’s the meaning of providing such experiments without giving attempted explanation or theoretical supports.

[1] Digvijay Boob and Cristóbal Guzmán. Optimal algorithms for differentially private stochastic monotone variational inequalities and saddle-point problems.


**Q7 Justification For Your Score:**

Please refer to Q5.

**Q9 Complying With Reviewing Instructions:**

1: Yes.

---

### Decision · Program_Chairs · 2022-05-15

**Decision:**

Accept (Poster)

**Comment:**

Meta Review: Reviewers are in agreement that this is a strong contribution and the authors have addressed major concerns in their response